# Failure and Degradation Mechanisms of Steel Pipelines: Analysis and Development of Effective Preventive Strategies

**DOI:** 10.3390/ma18010134

**Published:** 2024-12-31

**Authors:** Marcin Kowalczyk, Jakub Andruszko, Paweł Stefanek, Robert Mazur

**Affiliations:** 1Department of Machine Design and Testing, Wroclaw University of Science and Technology, Ignacego Lukasiewicza 7/9, 50-371 Wroclaw, Poland; marcin.kowalczyk@pwr.edu.pl; 2Hydrotechnical Unit, KGHM Polska Miedź S.A., Polkowicka 52, 59-305 Rudna, Poland; pawel.stefanek@kghm.pl (P.S.); robert.mazur@kghm.pl (R.M.)

**Keywords:** selective corrosion, erosion, steel pipeline degradation, welds, strength analysis, finite element method (FEM), preventive strategies

## Abstract

The increasing challenges related to the reliability and durability of steel pipeline infrastructure necessitate a detailed understanding of degradation and failure mechanisms. This study focuses on selective corrosion and erosion as critical factors, analyzing their impact on pipeline integrity using advanced methods, including macroscopic analysis, corrosion testing, microscopic examination, tensile strength testing, and finite element method (FEM) modeling. Selective corrosion in the heat-affected zones (HAZs) of longitudinal welds was identified as the dominant degradation mechanism, with pit depths reaching up to 6 mm, leading to tensile strength reductions of 30%. FEM analysis showed that material loss exceeding 8 mm in weld areas under standard operating pressure (16 bar) induces critical stress levels, risking pipeline failure. Erosion was found to exacerbate selective corrosion, accelerating degradation in high-stress zones. Practical recommendations include the use of corrosion-resistant materials, such as duplex steels, and implementing integrated monitoring strategies combining non-destructive testing with FEM-based predictive modeling. These insights contribute to developing robust preventive measures to ensure the safety and longevity of pipeline infrastructure.

## 1. Introduction

Steel pipelines are fundamental to contemporary transportation systems, enabling the transfer of resources such as crude oil, natural gas, potable water, and chemicals. These pipelines form the backbone of the global economy by providing essential services. However, they are subjected to degradation and failures caused by various threats, among which the most significant are corrosion and erosion. Corrosion is a chemical and electrochemical process that deteriorates steel in aggressive environments, while erosion, induced by the flow of media containing abrasive particles, gradually wears down the material and weakens the pipeline structure.

Corrosion manifests in various forms, each presenting specific challenges. Particularly problematic are selective and localized corrosion, known for their difficult-to-detect nature, making them causes of sudden and unexpected failures [1,2]. These forms of corrosion develop deep within the material or in hard-to-access areas, significantly complicating early detection and assessment.

Selective and localized corrosion are often triggered by interactions with CO_2_, H_2_S, or water, leading to material loss and eventual pipeline cracking [3,4]. These issues are exacerbated by microbial activity and deposits, especially in water injection pipelines [5,6]. The development of selective corrosion depends primarily on the material composition, environmental factors, and electrochemical interactions. Environmental conditions, such as the presence of hydrocarbons and carbonates, affect the stability of passive layers on steel surfaces, increasing the risk of damage [7]. 

Localized corrosion is influenced by factors such as defect geometry, galvanic effects, and specific environmental conditions (e.g., CO_2_, H_2_S presence). These factors reduce pressure resistance and accelerate structural degradation [8,9,10,11]. Welds are a critical component of the system, particularly vulnerable to corrosion due to microstructural differences, mechanical stresses, and environmental factors [12,13,14]. The heat-affected zone (HAZ) is especially prone to degradation. Irregularities in wall thickness and geometric discontinuities in welded joints intensify corrosion processes. Long-term exposure to corrosive environments, particularly those containing chloride ions, increases the risk of stress corrosion cracking (SCC) [15,16]. 

This cracking is primarily influenced by environmental conditions, material properties, and electrochemical protection strategies. Anodic dissolution mechanisms and hydrogen embrittlement further complicate the effectiveness of cathodic protection. Additionally, buried pipelines are significantly affected by soil conditions [17,18,19,20].

Erosion compounds the issue by accelerating structural degradation when combined with corrosion. Solid particles in a corrosive environment pose a substantial threat to pipeline integrity [21,22,23,24,25,26]. Additional threats, such as hydraulic impacts, mechanical damage, atmospheric phenomena, and environmental interactions, further complicate the situation [27,28]. These generate pressure waves and cyclic stresses, leading to material fatigue.

Given these processes, early detection of corrosion in steel pipelines is critical to ensuring their reliability and longevity. While visual inspections are commonly used, corrosion processes often require advanced techniques such as ultrasound, radiography, or thermography [29,30]. Environmental control does not fully eliminate risk, and the economic, environmental, and social consequences can be significant [31,32].

One key approach to preventing failures is eliminating corrosion from the analyzed environment. A first step might involve altering the chemical composition of the steel. Adding alloying elements such as chromium improves resistance to corrosion, particularly selective corrosion. This underscores the importance of selecting appropriate materials and implementing effective protective strategies during pipeline design and operation [7,33]. Mechanistic models predicting corrosion, accounting for microbiological activity, have also been developed [10].

When altering the chemical composition of steel is not feasible, other types of steel should be considered. Proper material selection minimizes corrosion risks. While carbon steel is inexpensive and widely used, it proves insufficient in highly corrosive or erosive conditions [34]. In such cases, low-alloy or stainless steels with high chromium content are preferred, as they form protective passive layers. Alternatively, duplex steels combine high strength with corrosion resistance. Corten steel, containing copper, chromium, or nickel, forms a durable oxide layer but requires detailed analysis before application [35,36,37].

The quality of welds plays a key role minimizing failure risks. Welding defects can initiate fatigue-related damage. Prolonged operation leads to degradation of internal surfaces, reducing fatigue strength and system reliability [38,39,40,41,42]. Ensuring high weld quality is therefore essential.

Underground pipelines face threats from seismic activity, ground structure disturbances, and rock mass movements. Despite extensive literature on these topics, there remains a lack of detailed guidelines for monitoring pipeline condition and early interventions, such as replacing potentially faulty sections [43,44,45]. Effective risk management requires systematic inspections, regular maintenance, and timely upgrades. Corrosion and erosion are interconnected threats, highlighting the need for comprehensive protective strategies [46]. Proper monitoring and preventive measures are crucial for maintaining the reliability of steel pipeline systems.

Addressing the challenges of degradation and failure in steel pipelines requires advanced research and practical solutions. This study focuses on selective corrosion and erosion as the primary degradation mechanisms. By combining experimental testing with numerical simulations, this analysis aims to provide a comprehensive understanding of these issues. The findings are intended to inform the development of effective preventive strategies and optimize the design and operation of pipeline systems.

This research combines experimental analysis and numerical modeling to address these challenges, providing a comprehensive understanding of degradation mechanisms. The findings support the development of practical strategies for improving pipeline safety, selecting optimal materials, and implementing advanced monitoring systems, ensuring reliable and cost-effective operations. Detailed methods and findings are discussed in subsequent sections, with Section 2 focusing on the study objects and research methodology.

## 2. Methodology and Objects of Study

Research on steel pipeline failures focuses on identifying the causes and mechanisms of degradation. Various testing methods and models are used to understand the structural issues leading to failures. This article analyzes the causes of pipeline degradation and presents strategies to prevent failures.

Comprehensive research was conducted to investigate these mechanisms. Experimental analyses were combined with numerical modeling to assess damage causes and prevention methods. This study evaluated the quality of welded joints and conducted strength simulations.

### 2.1. Description of the Study Objects and Sample Preparation

The study focused on steel pipelines used in the transport of mine water, process water, and flotation waste. These pipelines, made primarily of L235-grade steel with longitudinal SAWL welds, have diameters of 800 mm and 1000 mm. Significant failures included longitudinal seam ruptures and point leakages, which were attributed to corrosion, erosion, and mechanical damage. Failures in these pipelines pose significant environmental risks and were analyzed in detail in the following sections.

An analysis revealed varied failure cases, which are illustrated in Figure 1. To investigate these failures, a systematic data collection approach was implemented. Data were collected through field inspections and analysis of technical documentation of pipelines. Samples from 26 pipeline sections were taken for further laboratory analyses, focusing on welded joints and areas affected by selective corrosion and erosion.

Figure 2 shows an example of a damaged pipeline segment resulting from the longitudinal rupture of the seam.

Visible defects in the weld caused by selective corrosion are characterized and shown in Figure 3.

It is important to note, especially in the case of the unruptured internal section, that the nature of the acting selective corrosion is evident (Figure 3). Along the weld face, localized pits in the structure are visible, with significant longitudinal pits forming near the heat-affected zone (HAZ), which considerably reduce the cross-sectional area of the weld. 

Due to availability, for comparison with failed pipes, an analysis was also conducted on pipes that, due to their level of degradation, were replaced before failure. A characteristic feature of these pipes is the presence of hard and absorbent corrosion product coatings. This issue is illustrated in Figure 4.

After the removal of the corrosion products, this pipe also exhibits, in this case due to its earlier replacement, the initiation of degradation through the mechanism of selective corrosion. Longitudinal pits are visible near the HAZ, causing significant weakening of the weld cross-sectional area, as shown in Figure 5.

Within the group of pipes that did not fail, one was identified as atypical for the entire system, as it features a spiral SAWH seam, unlike the others. Despite the absence of cracking, selective corrosion was found along the weld face on the interior side after cleaning it of corrosion products, as well as pits along the entire length near the HAZ of the weld, as shown in Figure 6.

### 2.2. Wear Mechanisms of Pipelines

Based on the presented examples from the analyzed pipelines, two groups of pipelines differing in purpose were distinguished due to the wear mechanism.

A.For water transportation;B.For the transportation of flotation waste.

The distinguished types of pipeline exhibit different properties despite using the same materials. Water transport pipelines are not subject to intense abrasive wear. The most significant cause of failure is corrosion, particularly selective corrosion, which mainly affects longitudinal seams. These pipelines develop a compact, absorbent, and yet permeable layer of corrosion products on the interior. They are extremely difficult to diagnose due to internal encrustation, the nature of degradation, and the lack of easy access to the elements exposed to them. 

The schematic of this type of corrosion formation in the analyzed case is shown in Figure 7. This phenomenon is initiated by the retention of water (2) between the pipe material (1) and the compact, semipermeable layer of corrosion products and contaminants (3), where a saturation zone of corrosion products forms. The concentration of corrosion ions in this zone exceeds the saturation point, leading to localized precipitation of corrosion products and the formation of a water-metal interface.

On the external side of the pipe, there is a protective layer (4) designed to protect the material from external factors. When this layer is present, the selective corrosion mechanisms are limited to the internal surface of the pipe, where the absence of a protective coating allows direct contact between the corrosive medium and the metal.

In the pipe structure, which is not coated on the internal side, circumferential stresses (σr) occur. These stresses result from differences in the stress distribution between the inner and outer sides of the pipe, amplified by the pressure of the water network (p) acting from within the pipe. These stresses contribute to variations in the corrosion reaction rate on the internal surface of the pipe.

The development of selective corrosion along the HAZ (5) is particularly intense in areas with the highest stress gradient combined with high network pressure. Here, pits (6) preferentially form: deep material damage resulting from locally intensified corrosion processes.

### 2.3. Operating Conditions of the Pipelines

To properly diagnose operational problems and develop effective failure prevention strategies for steel pipelines, it is essential to understand the working conditions of such a system. It has been identified that, under standard operating conditions, the pressure in the pipes does not exceed 16 bar and observed pressure fluctuations do not exceed 50% of the baseline value. During testing, newly constructed pipelines are subjected to pressure tests at 25 bar, while new manufacturers’ pipes are tested at even higher pressures, such as 55 bar. 

Surface pipeline installations use compensators, which allow for the accommodation of length variations in the structure due to various factors, including temperature changes, which can reach several hundred millimeters. In cases where the pipe is near a compensator, a special support is used that not only accommodates axial displacements of the pipe but also allows the pipe to rotate relative to the support. At the support point, friction forces and loads resulting from misaligned support occur. For pipeline sections located underground, no additional supports are used, which is a standard industry practice. Exceptions are made in emergency situations where they are implemented as a remedial measure to ensure the continued safe operation of the system.

From these observations, it follows that accurately characterizing the mechanisms causing pipeline failures, as well as properly understanding their operational characteristics, can lead to the development of original methods to determine the causes of failures and create effective preventive strategies. The authors, addressing this issue, conducted studies on steel pipe fragments to identify potential causes of degradation and failure in steel pipelines and to evaluate the possibilities for preventing these phenomena. 

### 2.4. Research Methods

The basis for analyzing the complex problem of selective corrosion was the assessment of the level of degradation of the welded joints. To achieve this goal, metallographic sections were prepared in cross sections of the longitudinal welds, enabling a detailed analysis of the material structure. Additionally, to identify the causes of failure, studies were carried out on damaged pipeline sections, focusing on the quality of the joints. 

Corrosion analysis included testing in a ferroxyl reagent, consistent with ISO 8044 [47] for the classification of corrosion phenomena and ASTM G48 [48] for localized corrosion resistance in environments simulating chloride exposure. Complementary potentiometric measurements were performed on the areas of the welded joints to assess the electronic properties of the welds and the electrochemical potential affecting the corrosion processes. Selected pipeline sections were also subjected to microscopic examinations, allowing for a precise investigation of the nature and morphology of the pits formed in the heat-affected zones (HAZs) of the longitudinal welds. 

Subsequently, tensile tests were conducted on selected pipeline segments. The purpose of these tests was to determine the actual strength parameters of the materials used, quantifying the impact of degradation on the mechanical properties of the pipes. The final stage of the research involved non-linear strength analyses using the finite element method (FEM) for the welded joints of the pipelines, considering different levels of degradation. These analyses enabled computer modeling of the stress state in welded joints and prediction of their response to operational loads in the context of advanced corrosion processes.

## 3. Results

As a result of analyzing the degraded pipeline fragments, an attempt was made to investigate the causes of their degradation and the failure to develop effective preventive strategies. A series of studies was conducted on the degraded pipe fragments, as described in the following subsections.

### 3.1. Evaluation of the Degradation Level of Welded Joints

The level of degradation of the welded joints was assessed based on visual inspections and macroscopic examinations. Figure 8 shows a segment of the pipe after failure with a ruptured longitudinal weld, where the arrows indicate the locations with material loss in the weld.

The segment of the pipe shown in Figure 8 reveals varying degrees of corrosion loss along the weld on the internal side of the pipe. To evaluate and determine the extent of weld degradation, a segment of the same pipe that had not yet fractured was selected, as shown in Figure 9.

The selected segment (Figure 9) was divided to perform a macroscopic analysis of the welded joint. The selected fragments are shown in Figure 10.

The degree of weld degradation in the selected samples is shown in the macrographs (Figure 11) of the metallographic sections for the example segments. 

### 3.2. Assessment of Weld Quality in Damaged Pipeline Segments

To confirm the location of anodic areas in the welded joints of the pipes, corrosion tests were conducted on the joints described previously. The evaluation adhered to the requirements of ISO 5817 [49], which defines acceptable levels of weld imperfections, ensuring objective assessment criteria. Additionally, the qualifications of welders performing the joints were verified according to ISO 9606-1 [50], aligning with industry standards.

The corrosion test aimed to identify areas where oxidation (anode) and reduction (cathode) reactions occur. The test was performed using a ferroxyl reagent. The test was conducted using the ferroxyl reagent prepared with 100 cm^3^ of a 3% NaCl solution, 1 cm^3^ of a 1% K_3_[Fe (CN)_6_] solution, and 1 cm^3^ of a 1% phenolphthalein solution. The results of the tests, for example, welds with the highest degree of degradation, are shown in Figure 12.

These tests were qualitative in nature and provided only an approximate visualization of the areas where corrosion cells form. The results obtained confirmed the assumptions about the location, specifically the weld and its heat-affected zone (HAZ).

Quantitative studies of the corrosivity of welded pipe joints were also conducted to determine potential differences between weld areas, the HAZ, and pipe material. Based on the results of the potentiometric test, the susceptibility to galvanic corrosion can be assessed in or near the weld area. The designations of the joint samples subjected to potentiometric measurements are summarized in Table 1. 

In addition to joint samples cut directly from the pipe, comparative test joints were made using the steel from the examined pipe, welded with an acidic electrode (with a chemical composition like the pipeline steel) and an electrode with COR-TEN steel filler metal. The chemical compositions of the electrodes used are provided in Table 2.

Electrochemical studies were conducted in a three-electrode system: the working electrode—sample, the reference electrode—saturated calomel electrode, and the counter electrode—platinum wire. The measurements were conducted at a controlled temperature of 20–25 °C in a 3% NaCl solution to simulate a saline environment. The potential range was set from −1.5 V to +1.5 V relative to the reference electrode, with a scanning rate of 5 mV/s. 

Samples were stabilized in the electrolyte for 10 min before measurements, and IR compensation was applied to minimize resistance effects. The studies were performed in three areas of the sample: the weld area, the heat affected zone (HAZ), and the base material. Potential changes were recorded over a period of 1000 s.

Figure 13 present the results of potential recording studies over time. Each graph shows the potential curves for four different types of samples.

Table 3 presents the results of the electrochemical study, showing the potential values of the tested samples in three areas at 1000 s.

### 3.3. Microscopic Examination of Pitting in the Heat Affected Zone of the Welded Joints

Microscopic examinations were conducted on specimens in the initial etched state, observing cross-sections of previously selected samples. A Leica M205 stereomicroscope (Leica Microsystems, Wetzlar, Germany) was used for the study. The results of microscopic observations of pitting in the heat-affected zone (HAZ) based on examples of sections (Figure 11) are presented in Figure 14.

### 3.4. Strength Testing of Pipelines and Welded Joints

A static tensile test at room temperature was performed according to the PN-EN ISO 6892-1:2016-09 standard [51], method B30. The test was carried out using a Zwick/Roell Z100 THW (ZwickRoell, Ulm, Germany) universal testing machine equipped with an automatic macroXtens II extensometer (ZwickRoell, Ulm, Germany).

Several types of samples were selected for the tests. Material was taken from the base structure of the pipe (Figure 10) transversely to the axis of the welded joint, which shows the presence of corrosion pits (SPs). Figure 15 shows representative SP test specimens. The summary results of the mechanical properties for each test are presented in Table 4.

Figure 16 presents examples of tensile curves obtained during tests for SP samples, which, due to the nature of the existing pits, play a crucial role in the assessment method for determining the causes of pipeline failures.

### 3.5. Finite Element Method (FEM) Strength Analysis of Degraded Welded Joints

One of the research topics was to determine the levels of stress and plastic strain in areas of selective corrosion in longitudinal welds. To assess the impact of weld degradation on the strength of the welded joint in the pipeline, preliminary numerical calculations were conducted using the finite element method (FEM) in alignment with recognized international guidelines for structural evaluation. 

The first stage involved the construction of a geometric model based on the technical documentation of the analyzed pipeline. Due to the use of non-linear calculations, a symmetrical model was adopted for a 60° section of the pipe. 

Figure 17 shows the schematic used to model the pipe section and the example dimensions of the model.

In the non-linear numerical calculations, the maximum operating pressure in the pipeline was set at 16 bar. The numerical simulations were performed in multiple stages, gradually reducing the weld thickness by increments of 1 mm from the initial thickness, until half of the weld was removed.

The results, presented as contour graphs of circumferential normal stresses and equivalent plastic strains, are shown for various representative levels of weld degradation in Figure 18 and Figure 19.

Subsequently, models were developed to more accurately represent the shape of the pit formed along the weld due to corrosion (Figure 20), and a new series of non-linear calculations was performed, achieving a higher level of maximum pressure.

The objective of these calculations was to determine the allowable degradation of the pipe due to selective corrosion, and if it affects the longitudinal weld.

The example results of the numerical analyses, presented as contour plots of circumferential normal stresses and equivalent plastic strains at a pressure of 30 bar and a pit depth of 6 mm, are shown in Figure 21.

Figure 22 illustrates the impact of pressure and the degree of joint degradation on the plastic strain at the bottom of the notch in the hole caused by selective corrosion.

## 4. Summary and Discussion

The authors present a comprehensive analysis of the causes of industrial steel pipeline degradation and failure, focusing on damage mechanisms and methods for their prevention. Various research methods were applied in the study. The initial step was to determine the degree of degradation of the welded joints in selected areas of the steel pipeline. From the macrostructures shown in Figure 11, it is evident that corrosion in the case of welded joints is initiated primarily in the heat-affected zone (HAZ) near the internal weld of the pipe, first along the fusion boundary, and over time also within the internal weld itself. This is caused by the action of selective corrosion of the HAZ and selective corrosion of the weld metal. Additionally, localized pitting corrosion occurs in the weld, penetrating the weld metal. As a result of deep pit penetration, selective corrosion extends into the HAZ from the external weld of the pipe. In some cases, the weld penetrates the entire thickness of the joint. 

The corrosive environment played a significant role in accelerating degradation, particularly in the presence of CO_2_ and H_2_S. High chloride ion concentrations further intensified selective corrosion processes, aligning with findings in the literature on carbon steel corrosion in saline and acidic environments. Chemical analysis of corrosion products revealed semi-permeable layers on the inner pipe surfaces, retaining water and facilitating localized pitting. The chemical composition and microstructure play a critical role in the degradation mechanisms of steel pipelines. Alloying elements such as chromium and nickel enhance the formation of stable passive layers, mitigating susceptibility to selective corrosion, particularly in chloride-rich environments. 

The observed differences in electrochemical potentials between the base material and the weld metal indicate the necessity of optimizing electrode composition during welding. Furthermore, microstructural analysis of the heat-affected zones (HAZs) revealed residual stresses that intensify localized corrosion. Controlling welding parameters and selecting corrosion-resistant alloys could significantly reduce degradation in these critical areas. Analysis of the degree of degradation of the welded joint in pipes transporting water indicated the importance of selecting an appropriate filler metal for welding transmission pipelines, one that would change the potential distribution in the weld joint areas so that the weld metal does not become the anode. 

Consequently, potentiometric measurements were performed on selected welded joints from pipes and joints made with specific filler metals to investigate the potential distribution of the weld metal, the HAZ, and the base material. For these types of measurements, in most cases, the characteristic shape of the curves for the three types of area relative to a particular sample was similar. However, differences were observed between the various types of samples. In some cases, a small initial increase in the potential difference was observed at the beginning of the measurement, followed by a slow decline. No significant differences were recorded in the final potential values. In extreme cases, the potential difference, depending on the type of area, ranged from 20 to 40 mV. 

Potentiometric studies suggest that the weld metal is the anodic area most often, but in some instances, the base material can become the anodic area, which would be a more favorable configuration due to the surface area differences between the two regions of the welded joint. These findings extend upon earlier studies ([52,53]), which primarily focused on the anodic behavior of the HAZ without addressing local differences in susceptibility to corrosion along the length of the weld. In this study, the authors detailed how selective corrosion progresses asymmetrically along the weld, influenced by thermal effects and residual stresses. Additionally, the phenomenon of “tramline corrosion”, a symmetrical distribution of corrosion pits along the weld, was identified and linked to variations in welding parameters, a novel observation not previously detailed in the literature.

Microscopic analysis of pits in the heat-affected zone (HAZ) revealed that a common feature for many observed cross sections of both longitudinal and circumferential welds is the presence of selective corrosion in the HAZ. As a result, the resulting corrosion pits are symmetrically distributed around the weld, forming pairs of longitudinal grooves along the welded joint, a typical characteristic of corrosion known as tramline corrosion. The appearance and depth of the pits vary between different pipelines. In some cases, even along the length of a single welded joint, the morphology and depth of the pits change. The pits visible on the cross sections, penetrating the material along the fusion line and reflecting the shape of the weld, best illustrate the selective nature of the corrosion. 

Fatigue corrosion was identified as a significant mechanism contributing to pipeline failure. Pits formed in the HAZ acted as stress concentrators, initiating cracks under cyclic loading. Additionally, hydrogen-rich environments increased susceptibility to embrittlement, potentially undermining the effectiveness of cathodic protection systems. Furthermore, the synergistic impact of erosion and corrosion, a topic widely recognized in the literature [21,22,23,24,25,26], was expanded in this study through the practical observation of erosion-initiated pits that acted as focal points for accelerated corrosion. This finding highlights the dual threat of erosion/corrosion mechanisms in specific operating environments, such as pipelines carrying slurry or abrasive materials.

A crucial aspect of the study was the strength testing, which was performed using tensile testing. Based on the tensile tests performed, it was determined that the strength parameters obtained, along with the expanded measurement uncertainty, correspond to the structural steel grade S235 (JR, J0, J2) according to the PN-EN 10025-2:2004 standard [54]. The welds (SSH samples) exhibited a higher yield strength and tensile strength than the corresponding base materials of the pipelines. However, the pits formed due to corrosion processes acted as stress concentrators and served as initiation points for cracks that propagated along the boundary between the weld beads and the HAZ. This aligns with findings in earlier studies [16,17] regarding stress corrosion cracking (SCC), but this research goes further by employing FEM analyses to define critical degradation thresholds for SCC susceptibility.

Knowing the nature of joint degradation, the exact appearance of the resulting pits, and the material data collected from the analyzed pipe samples, finite element method (FEM) strength calculations were also performed. The strength properties of the material indicate a strain level of approximately 15% at which material rupture can occur. For a pipeline with a diameter of 1000 mm, the loss of thickness permitted is 8 mm, assuming a 10% plastic strain level. These calculations provide practical thresholds for allowable wear, a significant addition to the existing body of knowledge.

Thus, the findings from this study build upon and extend the existing literature, offering a nuanced understanding of degradation mechanisms in steel pipelines. By integrating empirical observations with advanced modeling techniques, the authors provide critical insights into selective corrosion, erosion, and SCC, emphasizing their combined impact on pipeline integrity. This work lays the foundation for improved diagnostic and preventive strategies, essential for the safe operation of industrial pipelines.

## 5. Conclusions

The conducted research has enabled a comprehensive understanding of damage mechanisms in steel pipelines, identifying selective corrosion and erosion as the primary factors contributing to their degradation. The study adhered to recognized standards, including PN-EN ISO 6892-1 [51], ISO 8044 [47], ASTM G48 [48], ISO 5817 [49], and ISO 9606-1 [50], ensuring the reliability, repeatability, and comparability of the obtained results. Physical analysis of damaged pipeline sections, corrosion tests, microscopic examinations, and tensile tests allowed for the precise identification of potential failure causes and the evaluation of preventive measures.

Specifically, the studies confirmed that selective corrosion, primarily initiated in the heat-affected zone (HAZ) near the internal weld of the pipe, leads to a reduction in tensile strength by approximately 30% and pit depths of up to 6 mm, as determined by microscopic examinations and FEM analyses. These findings emphasize the critical role of selective corrosion in weakening the load-bearing cross-section of the joint. The localized occurrence of pitting corrosion further accelerates this degradation process, and deep pit penetration leads to the propagation of selective corrosion to the external weld, weakening the load-bearing cross-section of the joint. 

Erosion, as the second major degradation mechanism, was found to exacerbate corrosion, particularly in high-stress zones. FEM simulations showed that material loss in weld areas exceeding 8 mm results in critical stress levels under standard operating pressures of 16 bar, significantly increasing the risk of pipeline failure. The analysis demonstrated a strong correlation between erosion and corrosion, emphasizing the necessity of a holistic approach to risk management.

Tensile tests revealed that the mechanical properties of the pipeline material comply with the expectations for L235-grade steel, indicating the appropriate quality of the material used for pipeline construction. However, in degraded areas affected by corrosion pits with depths up to 6 mm, a marked decrease in load-bearing capacity was observed. These pits acted as stress concentrators, initiating cracks that propagated along the heat-affected zone (HAZ) and significantly reducing the structural reliability of the welded joints. However, the observed damage to the welded joints, primarily along the fusion boundary, confirms that selective corrosion significantly reduces the strength of the welded joint. 

Numerical analyses conducted using the finite element method (FEM) quantified the degradation thresholds, identifying that plastic strain levels of 10% are reached when material loss at weld sites exceeds 8 mm. These results provide practical guidelines for allowable degradation levels before pipeline replacement or reinforcement becomes critical. Stress and strain modeling in areas affected by selective corrosion provided better insights into the impact of degradation on the structural integrity of pipelines. Cyclic pressure fluctuations within the transport system significantly contributed to the development of stress corrosion cracking (SCC). FEM analyses revealed that circumferential stresses (σ_r_) were critical in initiating cracks at locations weakened by corrosion pits.

The research findings provide practical recommendations for pipeline design, maintenance, and conservation. For instance, the application of duplex steels in highly exposed elements can mitigate the risk of selective corrosion, particularly in zones prone to pit formation. Regular inspections using non-destructive testing methods, such as ultrasonic and tomography techniques, are critical for detecting material loss exceeding the identified threshold of 8 mm, as indicated by FEM simulations. Specifically, the application of corrosion-resistant steels, such as duplex steels, in highly exposed elements of pipelines is suggested to mitigate the risk of selective corrosion. The selection of appropriate filler metals should be based on minimizing galvanic effects, and pipeline design should account for the electrochemical potential distribution and microstructural differences between the weld and the base material. 

Regular inspections using non-destructive testing methods, such as ultrasonics, tomography, and potentiometric measurements, are crucial for early damage detection. Furthermore, the application of protective coatings and corrosion inhibitors should be standard in critical areas of pipelines, particularly in weld zones and the HAZ. FEM analysis results highlight the need for precise determination of allowable wear levels and prediction of critical moments for pipeline component replacement. Stress modeling can also serve as a tool for optimizing pipeline geometry and wall thickness to improve structural integrity.

The author’s investigation into the causes of failure and degradation of steel pipelines provides valuable insights into key damage mechanisms and offers guidance for developing more effective preventive strategies. Understanding these processes is essential for ensuring the long-term durability and reliability of pipeline infrastructure, which is fundamental to environmental and economic safety. This work represents a significant contribution to the field of steel pipeline degradation research, emphasizing a comprehensive approach to addressing corrosion and erosion issues. The analyses conducted are an important step toward enhancing the safety and longevity of pipeline infrastructure, offering practical recommendations for preventing failures and mitigating their adverse effects. 

The authors underscore the importance of continued research in this area, which could further advance the development of effective diagnostic and protective methods for industrial pipelines. Future research should address these challenges by developing corrosion-resistant materials, such as duplex steels, optimized for demanding conditions. The integration of advanced monitoring systems with predictive modeling techniques, such as the FEM, will enhance the accuracy of degradation assessments and preventive strategies. Further studies should also focus on understanding the synergistic effects of erosion and corrosion in environments with abrasive particles, as well as optimizing welding processes to mitigate residual stresses and galvanic effects in welded joints. Establishing critical degradation thresholds for timely maintenance and developing standardized diagnostic guidelines would significantly improve the safety and reliability of pipeline systems.

## Figures and Tables

**Figure 1 materials-18-00134-f001:**
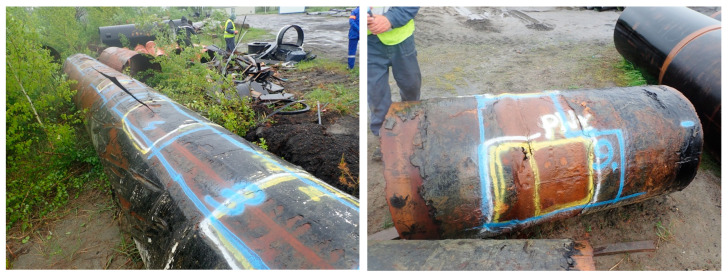
Example segments of the analyzed pipes.

**Figure 2 materials-18-00134-f002:**
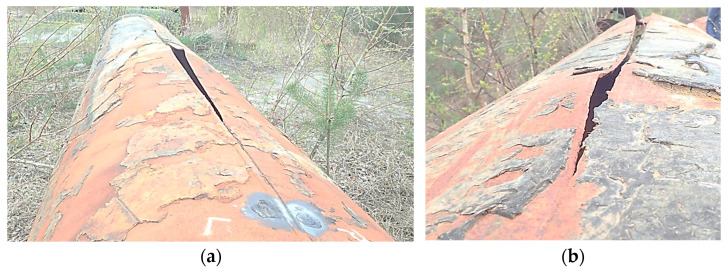
Example of a damaged water pipeline segment—longitudinal seam rupture: (**a**) General view along the crack and (**b**) close-up view of the seam rupture location.

**Figure 3 materials-18-00134-f003:**
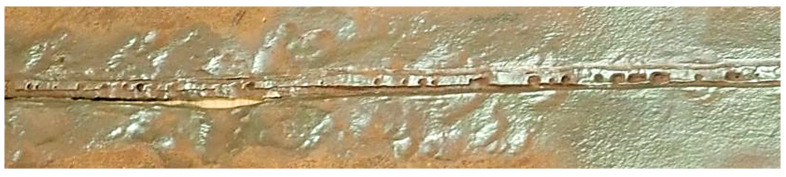
Weld from the inside of the pipe on the unruptured section after cleaning.

**Figure 4 materials-18-00134-f004:**
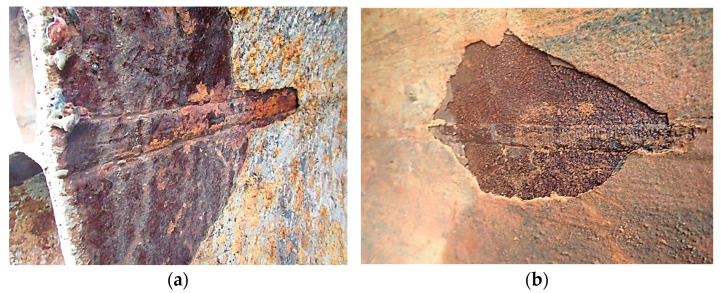
Hard, compact, subsurface permeable corrosion product coatings that separate the pipe material from the water flow inside the pipe: (**a**) Start of the pipe seam and (**b**) mechanical separation of the crust.

**Figure 5 materials-18-00134-f005:**
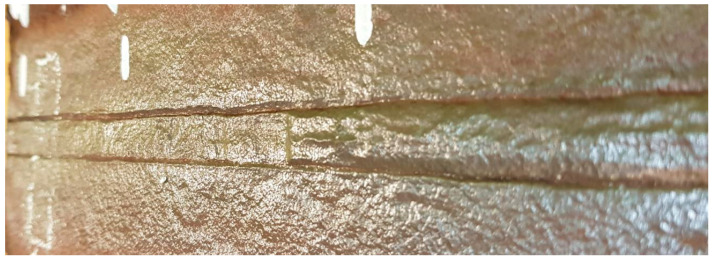
Pipe after the removal of corrosion products, showing characteristic pits that degrade the longitudinal weld of the pipe.

**Figure 6 materials-18-00134-f006:**
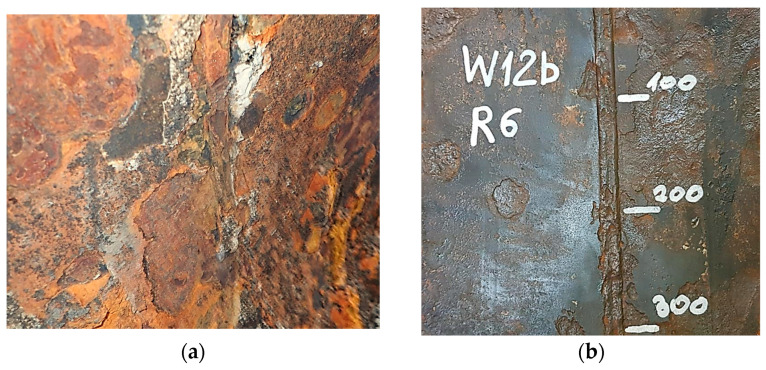
Pipe with a spiral SAWH seam: Longitudinal seam rupture in pipe P1, with visible defects in the weld: (**a**) Seam with visible corrosion products and (**b**) seam after cleaning off the corrosion products.

**Figure 7 materials-18-00134-f007:**
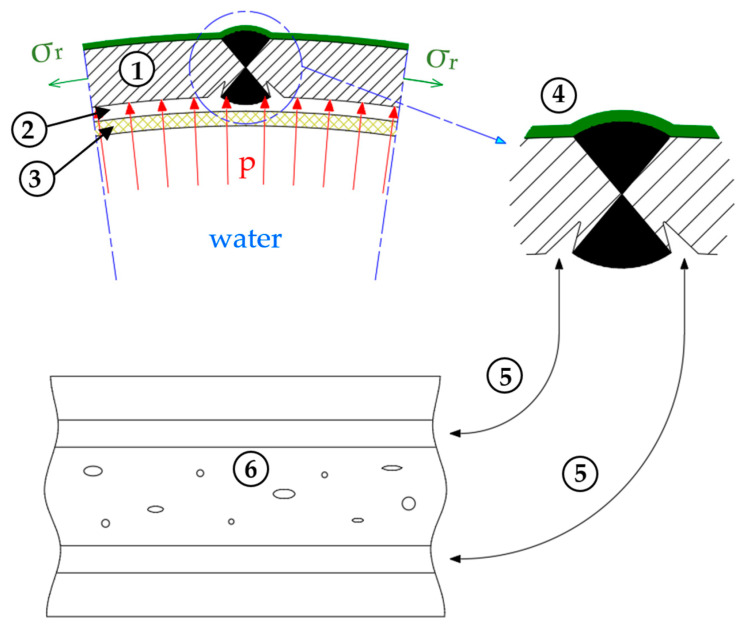
Water pipe with longitudinal seam—mechanism of selective corrosion formation: 1—steel pipe shell of the pipeline; 2—trapped water layer between the pipe material and the layer of corrosion products; 3—compact, semi-permeable layer of corrosion products and contaminants; 4—external protective layer; 5—areas of selective corrosion development along the HAZ of the longitudinal weld; 6—local pits as a result of selective corrosion. σr—circumferential stresses in the pipe; p—network pressure.

**Figure 8 materials-18-00134-f008:**
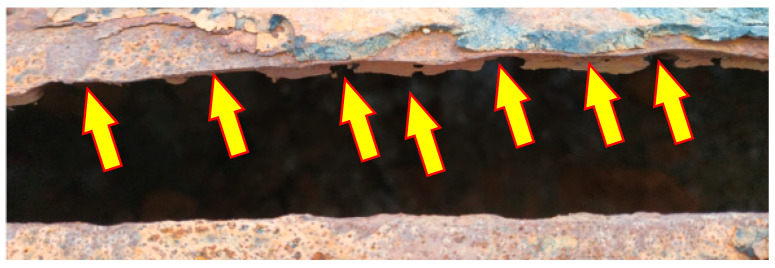
View of the remaining weld remnants at the edge of the joint with the longitudinal weld of the fractured pipe.

**Figure 9 materials-18-00134-f009:**
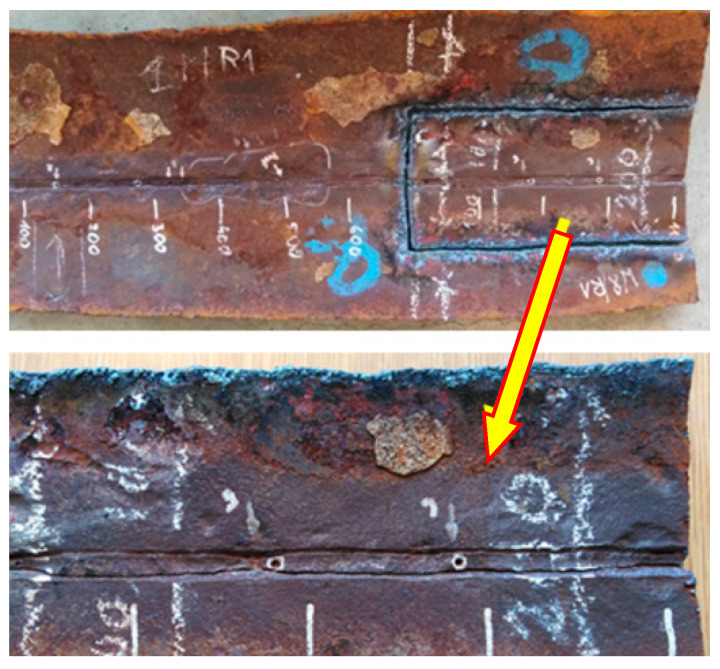
View of the internal surface of the pipe segment illustrating the degree of weld degradation for water transport pipes.

**Figure 10 materials-18-00134-f010:**
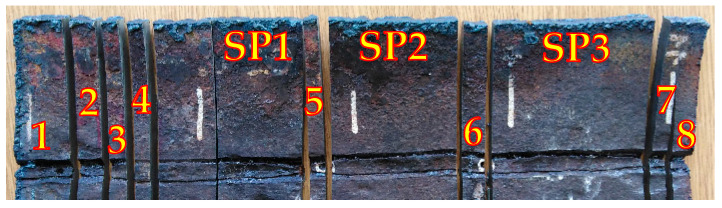
Division of the pipe segment into samples for macroscopic analysis (1–8) and mechanical testing (SP1–SP3).

**Figure 11 materials-18-00134-f011:**
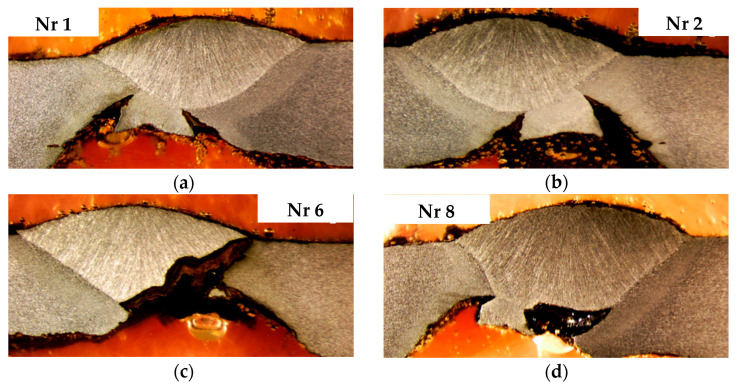
Macrostuctures of welded joint samples: (**a**) Sample 1, (**b**) Sample 2, (**c**) Sample 6, (**d**) Sample 8 from Figure 10.

**Figure 12 materials-18-00134-f012:**
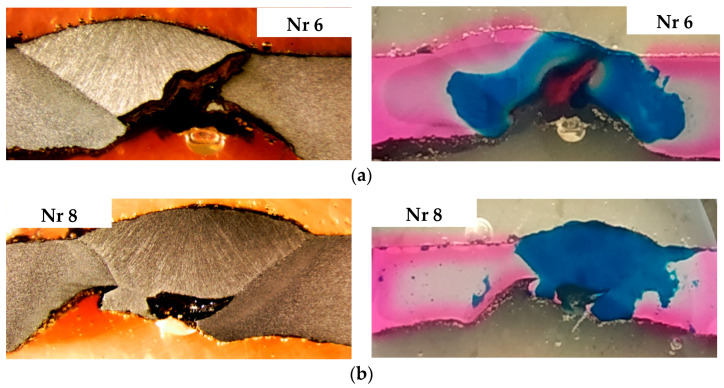
Anodic areas (**blue**) and cathodic areas (**pink**) in the welded joints of the pipe: (**a**) Sample 6, (**b**) Sample 8 from Figure 10.

**Figure 13 materials-18-00134-f013:**
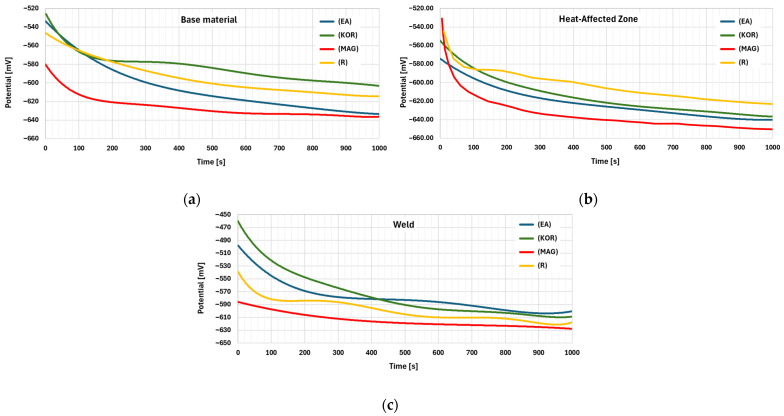
Potential profile: (**a**) Base material; (**b**) heat affected zone (HAZ) area; (**c**) weld area.

**Figure 14 materials-18-00134-f014:**
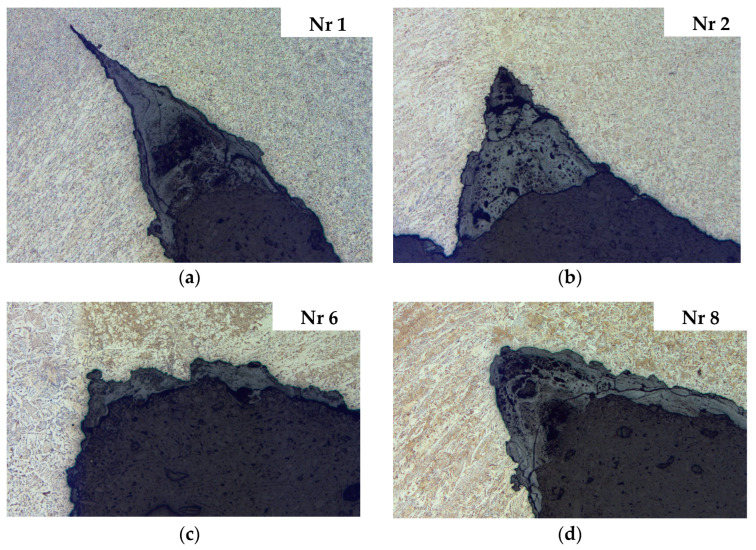
Microscopic analysis—Pits in HAZ: (**a**) Sample 1, (**b**) Sample 2, (**c**) Sample 6, (**d**) Sample 8 from Figure 10.

**Figure 15 materials-18-00134-f015:**
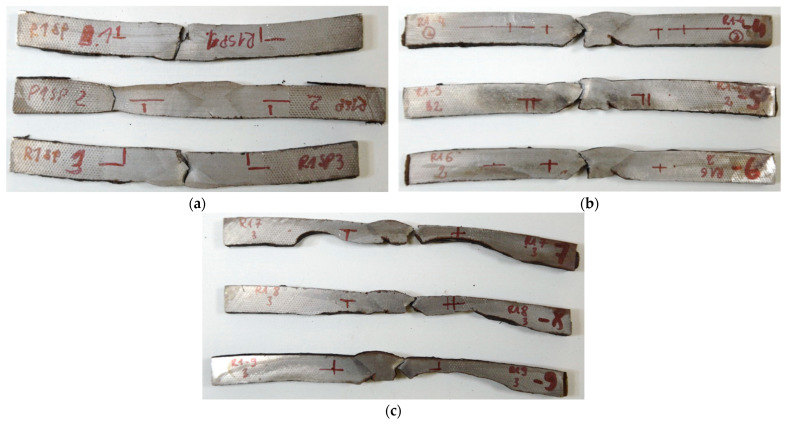
Samples after strength testing: (**a**)—SP1; (**b**)—SP2; (**c**)—SP3.

**Figure 16 materials-18-00134-f016:**
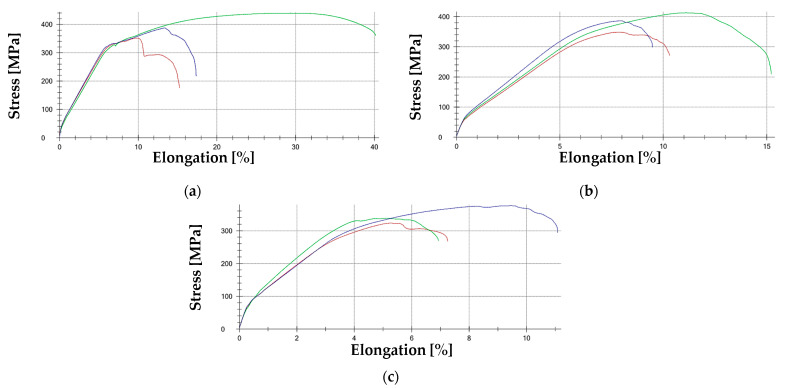
Tensile curves: (**a**)—SP1; (**b**)—SP2; (**c**)—SP3 where: green—Test 1, blue—Test 2, red—Test 3.

**Figure 17 materials-18-00134-f017:**
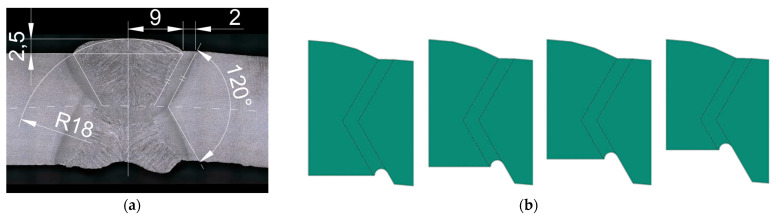
Schematics for modeling the pipe section and example model dimensions: (**a**) base weld cross-section dimensions; (**b**) schematic of weld material loss.

**Figure 18 materials-18-00134-f018:**
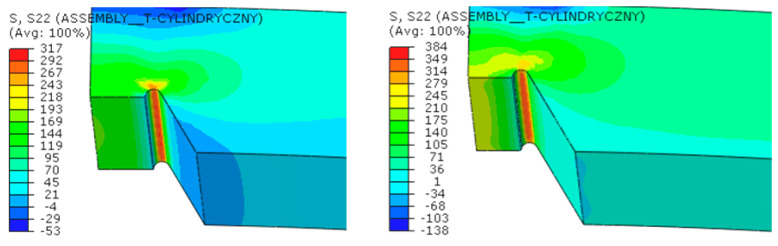
Circumferential normal stresses [MPa] at a pressure of 16 bar at various levels of longitudinal weld degradation.

**Figure 19 materials-18-00134-f019:**
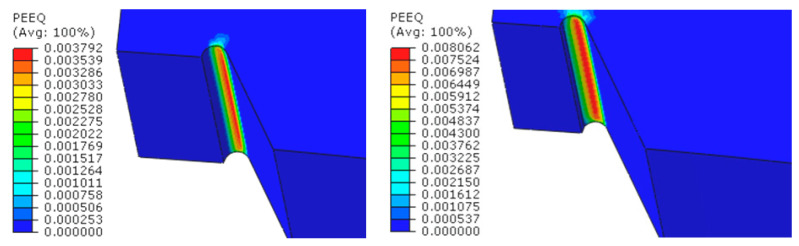
Equivalent plastic strains in the material at a pressure of 16 bar at various levels of longitudinal weld degradation.

**Figure 20 materials-18-00134-f020:**
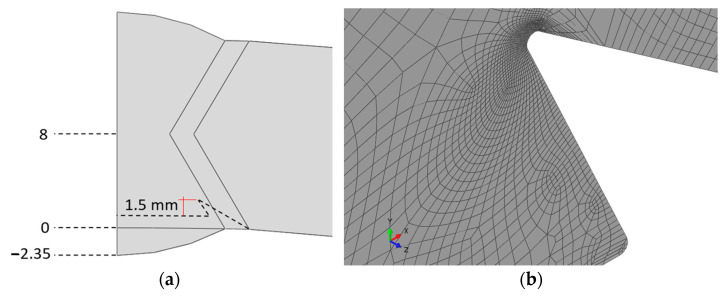
Sharp termination of the notch bottom in the pit caused by selective corrosion: (**a**) Dimensions of the model in the second stage of numerical analysis and (**b**) discrete model in the pit area during the second stage of numerical analysis.

**Figure 21 materials-18-00134-f021:**
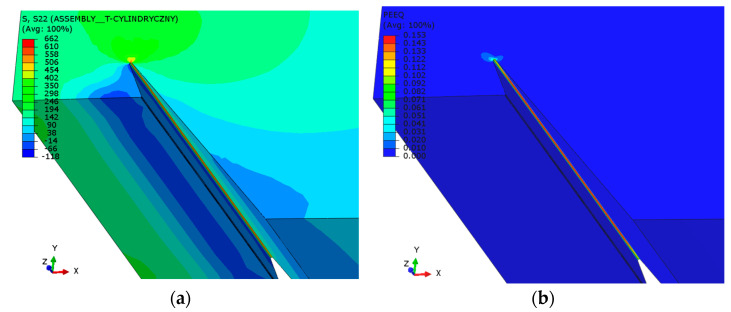
Results of the numerical analyses for the second stage: (**a**) Circumferential normal stress in [MPa] at a pressure of 30 bar and a degradation depth of 6 mm and (**b**) equivalent plastic strain at a pressure of 30 bar and a degradation depth of 6 mm.

**Figure 22 materials-18-00134-f022:**
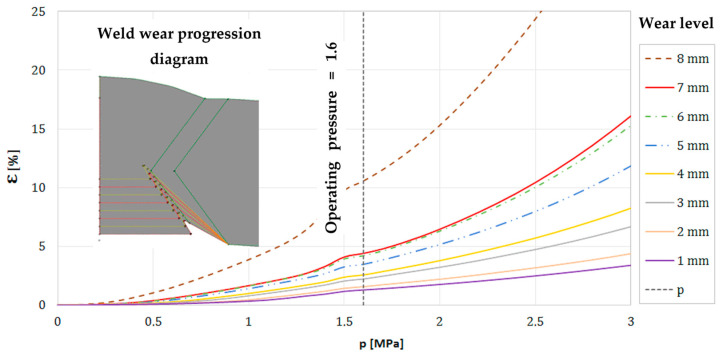
Plastic deformation as a function of pressure at the bottom of the hole in the longitudinal weld of the pipe depending on the depth of loss of material.

**Table 1 materials-18-00134-t001:** Designations of welded pipe joint samples.

Sample Identifier	Description
EA	Steel samples from the pipe welded with EA electrode
COR	Steel samples from the pipe welded with the COR-TEN electrode
MAG	Steel samples from the pipe welded using the MAG method
R	Original joint from R1 pipe

**Table 2 materials-18-00134-t002:** Chemical composition of weld metal from covered electrodes (EA and COR-TEN).

Electrode Type	Electrode Classification	Element Content in wt.%				
		C	Si	Mn	Ni	Cu
Pipe	Pipe	0.15	0.19	0.48	<0.01	0.013
Acidic	EA 146	0.07	0.1	1.0	-	-
Cor-ten	Tencord 74 Kb	0.06	0.40	1.0	1.0	0.45

**Table 3 materials-18-00134-t003:** Values of the steady-state potential of tested samples after 1000 s in process water.

Sample	Weld Area [mV]	HAZ [mV]	Base Material [mV]
EA	−604.91	−643.45	−635.09
COR	−609.54	−636.34	−603.11
MAG	−627.17	−650.37	−636.73
R	−619.43	−624.03	−615.86

**Table 4 materials-18-00134-t004:** Strength properties for individual tensile tests.

Sample	R_p0,2_	R_eH_	R_eL_	R_m_	F_m_	A_5,65_	L_0_	a_0_	b_0_	S_0_
	MPa	MPa	MPa	MPa	kN	%	mm	mm	mm	mm^2^
SP1—1	-	-	-	352	21.44	-	-	5.86	10.40	60.94
SP1—2	-	-	-	440	26.24	-	-	5.84	10.22	59.68
SP1—3	-	-	-	388	23.14	-	-	5.83	10.24	59.70
SP2—4	-	-	-	348	19.46	-	-	7.25	7.71	55.90
SP2—5	-	-	-	413	22.88	-	-	7.25	7.65	55.46
SP2—6	-	-	-	386	18.95	-	-	7.25	6.77	49.08
SP3—7	-	-	-	323	10.82	-	-	7.68	4.36	33.48
SP3—8	-	-	-	338	9.66	-	-	6.60	4.33	28.58
SP3—9	-	-	-	376	13.59	-	-	7.67	4.71	36.13

## Data Availability

The original contributions presented in this study are included in the article. Further inquiries can be directed to the corresponding author.

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
