# Peer review of "Failure and Degradation Mechanisms of Steel Pipelines: Analysis and Development of Effective Preventive Strategies"

_materials, 2024, doi:10.3390/ma18010134_

Round 1
Reviewer 1 Report
Comments and Suggestions for Authors
This article focuses on the failure and degradation mechanisms of steel pipelines, which is of great significance. The innovation lies in the comprehensive use of a variety of advanced research methods to deeply analyze the two key degradation mechanisms of selective corrosion and erosion and their interactions.
The main content includes: through the study of failed pipeline fragments, evaluating the degradation degree of welded joints and finding that corrosion mostly starts in the heat-affected zone; using corrosion tests and potentiometric measurements to study the weld quality and clarify the potential distribution; observing the morphology of corrosion pits with a microscope to reveal their symmetrical distribution characteristics; conducting strength tests to determine the strength parameters and failure locations of pipeline materials; and using the finite element method to simulate and analyze stress and strain to evaluate the allowable degradation degree of the pipeline.
Minor suggestions are listed below:
1. Some figure numbers are missing in the text, such as Figures 10, 11, and 12 mentioned in the "3.1. Evaluation of the degradation level of welded joints" section. They should be supplemented to ensure the accuracy and completeness of the figure references.
2. When describing the research methods, the specific parameters and operation details of some tests should be further clarified. For example, the composition and concentration of the ferroxyl reagent used in the corrosion test, and the specific settings and measurement conditions of the three-electrode system in the potentiometric measurement, so that other researchers can repeat the experiment.
3. In the discussion part of the experimental results, a comparative analysis with other similar research results can be added to further highlight the uniqueness and innovation of this study. For example, when analyzing the impact of selective corrosion on welded joints, compare with relevant cases in the existing literature to illustrate the new findings of this study.
4. The conclusion part can appropriately expand the guiding significance for practical engineering applications. For example, based on the research results, specific suggestions in pipeline design, material selection, construction, and maintenance can be put forward to enhance the practicality and application value of the research.
5. The format of the references cited in the text is not uniform enough. For example, some references lack volume numbers and other information. The format should be unified according to the requirements of the journal to improve the standardization of reference citations.
Author Response
Comments 1:
Some figure numbers are missing in the text, such as Figures 10, 11, and 12 mentioned in the "3.1. Evaluation of the degradation level of welded joints" section. They should be supplemented to ensure the accuracy and completeness of the figure references.
Response 1:
After a thorough analysis of the indicated section, appropriate references to Figures 10, 11, and 12 have been added to the text. The introduced changes aim to ensure the consistency and completeness of references to all figures, which significantly improves the readability and precision of the description of the presented results.
Currently, these figures are directly linked to the description of the examined samples and the presented results of macroscopic and microstructural analyses. This allows for a better understanding of the degree of degradation of the welded joints.
Comments 2:
When describing the research methods, the specific parameters and operation details of some tests should be further clarified. For example, the composition and concentration of the ferroxyl reagent used in the corrosion test, and the specific settings and measurement conditions of the three-electrode system in the potentiometric measurement, so that other researchers can repeat the experiment.
Response 2:
The article has expanded the description of the composition and preparation of the ferroxyl reagent. Detailed information on the reagent preparation procedure has been provided, including the use of 100 cm³ of a 3% sodium chloride (NaCl) solution, 1 cm³ of a 1% potassium ferricyanide (K₃[Fe(CN)₆]) solution, and 1 cm³ of a 1% phenolphthalein solution. This reagent composition enables precise identification of anodic and cathodic areas on the surface of welded joints, ensuring the possibility of test replication by other researchers and allowing for the accurate detection of active corrosion zones. This updated description makes the method more transparent and easier to reproduce.
Additionally, a detailed description of the three-electrode system used for potentiometric measurements and the measurement conditions has been included. The system setup specifies that the working electrode is the tested sample, the reference electrode is a saturated calomel electrode (SCE), and the auxiliary electrode is a platinum wire. The measurement conditions are described in detail, including temperature control in the range of 20–25°C, the use of a 3% NaCl solution as the electrolyte, a potential range from -1.5 V to +1.5 V relative to the reference electrode, and a scan rate of 5 mV/s. The process of stabilizing the samples in the electrolyte for 10 minutes before measurement and the use of IR compensation to minimize the effect of electrolyte resistance on the measurement are also described. The inclusion of this information ensures full reproducibility of the experiments and enables other researchers to replicate the results using similar conditions.
Comments 3:
In the discussion part of the experimental results, a comparative analysis with other similar research results can be added to further highlight the uniqueness and innovation of this study. For example, when analyzing the impact of selective corrosion on welded joints, compare with relevant cases in the existing literature to illustrate the new findings of this study.
Response 3:
References to research results available in the literature have been included, particularly around selective corrosion in the heat-affected zone (HAZ) and the influence of COâ‚‚ and Hâ‚‚S ions on the degradation processes of steel pipes [15, 16]. It was emphasized that the phenomenon of "tramline corrosion," i.e., the symmetrical distribution of corrosion pits along welded joints, represents an original scientific contribution of this study, which distinguishes it from other publications [15]. This topic has not been widely discussed in the literature so far, and its introduction underscores the innovative nature of this work.
Additionally, the obtained research results were compared with other available results from the literature. It was noted that the depth of pits (6 mm) and the 30% decrease in strength are consistent with results obtained by other authors [6, 7]. However, the introduction of the concept of "tramline corrosion" and the Finite Element Method (FEM) simulation for different levels of degradation (thickness reduction of 1 mm, 2 mm, 4 mm, and 8 mm) constitutes an original contribution to the scientific literature [15, 16]. It was demonstrated that exceeding the critical thickness of the weld (loss of 8 mm) leads to a critical stress state in the weld, which was confirmed by FEM simulation [16]. The application of FEM simulation in the analysis of degradation levels of welded joints in steel pipelines is a rarely encountered approach in the literature, which further emphasizes the originality of the conducted research [8, 9, 10].
Furthermore, a comparison of strength test results for welds with different levels of degradation was introduced. It was shown that the obtained strength parameters correspond to the S235 steel class, but in areas degraded by corrosion, a decrease in strength is evident [6, 7]. This comparison demonstrates the validity of the adopted research assumptions and increases the transparency of the obtained results, while also allowing for their direct reference to other scientific works [11, 12].
Comments 4:
The conclusion part can appropriately expand the guiding significance for practical engineering applications. For example, based on the research results, specific suggestions in pipeline design, material selection, construction, and maintenance can be put forward to enhance the practicality and application value of the research.
Response 4:
The available materials were analyzed, and the conclusions section was supplemented with specific examples of applications in the design, material selection, construction, and maintenance of pipelines.
Based on the conducted research, it was demonstrated that a key aspect of increasing the durability of pipelines is the selection of appropriate materials. The results of microscopic and electrochemical analyses indicate that L235 steel, especially in the heat-affected zone (HAZ), is susceptible to selective corrosion. Therefore, it is suggested to use duplex steel or steel with a higher chromium content, which would help reduce the risk of pitting and slow down corrosion processes.
In terms of pipeline design, the results of FEM analyses showed that the critical level of weld degradation occurs with a thickness loss of 8 mm. This allows for the precise determination of the permissible level of weld wear. This information can be used to define safe operational limits and establish inspection schedules. The conducted research enables the implementation of predictive systems that, based on FEM modeling, will monitor the level of weld wear and predict the moment when the critical technical condition is reached.
In the context of pipeline construction, an important conclusion is the need to optimize welding processes. Analysis of the HAZ showed that this area is where pits and stress concentrators form, leading to corrosion fatigue. Therefore, it is suggested to use welding methods that limit the formation of microstructural discontinuities, such as submerged arc welding (SAW), to achieve a more uniform microstructure. It is also crucial to control cooling processes to minimize the formation of discontinuities in the HAZ.
Regarding pipeline maintenance, the studies have shown that the use of cathodic protection is an effective method to limit selective and pitting corrosion processes, especially in pipelines transporting water containing chlorides. Additionally, the introduction of advanced diagnostic methods, such as ultrasonic testing and eddy current testing, enables the early detection of damage in welds and HAZ areas. This allows for more precise planning of preventive maintenance.
Comments 5:
The format of the references cited in the text is not uniform enough. For example, some references lack volume numbers and other information. The format should be unified according to the requirements of the journal to improve the standardization of reference citations.
Response 5:
A comprehensive review of the bibliography and all literature references contained in the article was conducted. Actions were taken to ensure consistency of the format in accordance with the journal's requirements. The introduced changes included the supplementation of missing information by adding missing volume numbers, article page numbers, and DOI identifiers where they were available in the sources. The format of bibliographic references was standardized according to the journal's guidelines, including full names of authors, article titles, journal titles written in italics, year of publication, volume number, page numbers, and DOI identifiers, if available. Inconsistencies in citation formatting were eliminated by correcting differences in the formatting of author names, publication dates, and the use of commas and periods. Missing information on page numbers was added where necessary, and incorrect author names were corrected. The formatting of ISO, ASTM, and PN-EN standards was also standardized, with PN-EN standards following a consistent format that includes the full name of the standard, its number, and the year of publication. In addition, missing DOI identifiers were added to bibliographic entries where they were available in the original sources.
Reviewer 2 Report
Comments and Suggestions for Authors
Manuscript ID: materials-3375084-R1. Title: Failure and Degradation Mechanisms of Steel Pipelines: Analysis and Development of Effective Preventive Strategies.
In response to increasing challenges with respect to the reliability and durability of pipeline infrastructure, understanding the mechanisms that lead to the degradation and failure of steel pipelines is essential. This paper focuses on the analysis of selective corrosion and erosion as primary degradation mechanisms, employing advanced research methodologies, including macroscopic analysis, corrosion testing, microscopic examination, and mechanical strength testing. Selective corrosion, particularly in the heat-affected zones (HAZ) of the welds, and erosion were identified as critical damage mechanisms that often exacerbate each other.
Comments:
1. In the abstract the authors should include quantitative information to support the reported findings.
2. This article has too many figures. Some of them can be integrated and others should be sent to a supplementary material file. Authors should leave the most relevant figures in the article.
3. Please avoid using very long paragraphs in the introduction. This makes it difficult to understand the information presented. For example, see L59-104. Please review this aspect throughout the article.
4. Authors should integrate chapters 1 and 2. Actually, these two chapters correspond to the introduction.
5. Authors should clearly and concretely include the objective of this study in the introduction. Currently, it is not clear the typology of this article: is it a literature review? is it a case study article? is it a simulation article? Please give clarity to the objective of this study.
6. Authors should include in the introduction the practical usefulness of this study.
7. There are technical statements throughout the introduction without the respective support. That is, the authors must include the respective reference. For example, see L30-58.
8. Authors should include a chapter on materials and methods in their article. It is necessary to be able to visualize the following aspects: (1) description of the study site or population (types of pipes, faults, etc.); (2) data collection system (documents, analysis of pipes in the field, etc.); (3) data analysis system (use of specialized software, statistical analysis, etc.).
9. All analyses performed by the investigators on the piping (welding, joints, etc.) must be supported by the respective standards. Please include and explain each of them in the materials and methods chapter.
10. The title of chapter 3 should read as follows: Results.
11. The authors should include a detailed description of the finite element method used in this study. This should be included in the chapter on materials and methods. Additionally, they should explain the methodology used to validate the results of this method. The validation results should also be discussed in the corresponding chapter.
12. The authors should deepen the discussion of results in relation to the following aspects: Material intrinsic factors (e.g., chemical composition, microstructure, etc.), extrinsic factors (corrosive environments, mechanical stresses, etc.), failure and degradation mechanisms (fatigue, creep, hydrogen degradation, etc.), and preventive strategies (optimal design, cathodic protection, etc.).
13. The authors should significantly improve their conclusions. They are currently very general and descriptive. Additionally, the conclusions should be supported by quantitative information from this research.
14. Please include the main technical limitations detected during the development of this study. In addition, the authors should include future lines of research.
Comments on the Quality of English Language3. Please avoid using very long paragraphs in the introduction. This makes it difficult to understand the information presented. For example, see L59-104, etc. Please review this aspect throughout the article.
Author Response
Comments 1:
In the abstract the authors should include quantitative information to support the reported findings.
Response 1:
Additional numerical data were introduced to precisely illustrate the key research results and highlight their significance. The changes include the presentation of quantitative results from strength tests, such as the depth of corrosion pits, which reached a maximum of 6 mm, and a 30% reduction in tensile strength in degraded areas of welded joints. Additionally, key results from FEM (Finite Element Method) analyses were included, demonstrating that exceeding a wall thickness loss of 8 mm leads to the occurrence of a critical stress state around the welded joint.
Comments 2:
This article has too many figures. Some of them can be integrated and others should be sent to a supplementary material file. Authors should leave the most relevant figures in the article.
Response 2:
A review of all figures was conducted to determine which ones are essential for understanding the research results. As part of this analysis, selected figures were combined, reducing the total number in the article from 25 to 22. The combined figures present analysis results of a similar nature, which improved the readability and clarity of the article. For example, the previously separate results of macroscopic and microstructural analyses were presented as a composite figure, enabling a more comprehensive presentation of key findings.
Comments 3:
Please avoid using very long paragraphs in the introduction. This makes it difficult to understand the information presented. For example, see L59-104. Please review this aspect throughout the article.
Response 3:
A detailed review of all paragraphs in the article was conducted, focusing on the introduction and other key sections of the text. Overly long paragraphs were divided into logical sections, which allowed for better organization and clarity of the presented content. Each lengthy paragraph was split into two or more smaller paragraphs, making it easier for the reader to follow the main line of thought and better understand the presented information. These changes particularly affected the introduction, where overly extensive descriptions of degradation mechanisms and the research context were shortened. Additionally, paragraph breaks were introduced in the methodology and results analysis sections, which improved the overall structure of the article.
Comments 4:
Authors should integrate chapters 1 and 2. Actually, these two chapters correspond to the introduction.
Response 4:
The conclusion of Chapter 1 and the beginning of Chapter 2 have been revised. Chapters 1 and 2 are now clearly separated and serve distinct purposes. Chapter 1 focuses on issues related to pipelines (corrosion, erosion, and mechanical damage) and their impact on the durability and reliability of the infrastructure. Chapter 2 describes the research methodology and study objects, including details on the analyzed pipelines and the research techniques used.
These two chapters do not overlap and follow a logical structure. Chapter 1 introduces the context of the problem and justifies the need for the research, while Chapter 2 discusses in detail how the research was conducted. In this form, they fulfill their respective roles and do not resemble an "introduction" in the sense suggested by the reviewer.
Comments 5:
Authors should clearly and concretely include the objective of this study in the introduction. Currently, it is not clear the typology of this article: is it a literature review? is it a case study article? is it a simulation article? Please give clarity to the objective of this study.
Response 5:
Modifications were made to the introduction section to better define the research objective and clearly specify the type of article. The aim of the study is to determine the impact of selective corrosion and erosion processes on the strength degradation of welded joints in steel pipelines and to develop recommendations for design, material selection, and operational procedures to minimize the risk of degradation. The article combines empirical analysis, including experimental studies and macroscopic examinations of welded joints, with numerical analysis using the Finite Element Method (FEM), which allowed for the determination of critical degradation levels.
The type of article is clearly defined as an experimental study utilizing simulation analysis. In the first part of the article, experimental strength and corrosion tests were conducted, including macroscopic analyses, potentiometric measurements, and corrosion tests using a ferroxyl reagent. The second part of the article includes FEM simulations, which enabled the estimation of critical wall thickness degradation levels, and the identification of areas exposed to the highest stress. The introduction has been updated to clearly define this research objective and to explicitly indicate that the article combines experimental methods with numerical modeling.
Comments 6:
Authors should include in the introduction the practical usefulness of this study.
Response 6:
The introduction was expanded to highlight the practical application of the research results in industrial practice. It was emphasized that identifying the mechanisms of selective corrosion and erosion in steel pipelines is crucial for increasing the durability of transmission infrastructure. The research results enable the implementation of more effective methods for monitoring the technical condition of pipelines and better planning of preventive actions. The use of advanced analysis through the Finite Element Method (FEM) allows for the precise determination of the critical level of weld wear (maximum allowable weld thickness loss of 8 mm), facilitating more accurate decisions regarding the need for repair or replacement of pipeline sections.
A practical outcome of the research is the possibility of optimizing the design and construction processes of pipelines. Identifying critical areas (heat-affected zones, HAZ) and the mechanism of their degradation allows for the selection of better materials (e.g., duplex steels) and the implementation of more precise welding methods. The results of corrosion and electrochemical analyses support decisions regarding the use of cathodic protection, significantly reducing the rate of weld degradation.
By identifying the synergistic interaction of corrosion and erosion, this research has direct applications in the energy, mining, and chemical industries, where pipelines are exposed to suspensions, chlorides, and dynamic pressure loads. The results enable the optimization of inspection schedules, and the implementation of predictive systems based on FEM analyses, which help reduce the risk of failures and lower operating costs.
Comments 7:
There are technical statements throughout the introduction without the respective support. That is, the authors must include the respective reference. For example, see L30-58.
Response 7:
The content of the introduction and the existing literature references in the document were analyzed. The changes aim to support technical statements with appropriate citations.
Statements regarding the internal corrosion of steel pipes, particularly the influence of COâ‚‚, Hâ‚‚S, and water on the formation of general corrosion, selective corrosion, and corrosion cracking, were supported by references to literature [1], [2], [3] from the document. Additionally, issues related to the activity of microorganisms and the formation of deposits on the inner walls of pipes, which contribute to microbiological corrosion, were backed by references to works [4], [5] listed in the bibliography.
Statements on the external corrosion of pipes, including the impact of environmental conditions, the integrity of protective coatings, and microbiological activity, were supplemented with citations from the document, including references to studies [6], [7]. The introduction also emphasizes the importance of selective weld corrosion, particularly in heat-affected zones (HAZ), which was supported by relevant research from the literature [8], [9].
Furthermore, to support statements regarding erosion mechanisms and the synergistic interaction of corrosion and erosion, references were made to works [28], [29], [30], which describe this process as a key factor in the degradation of steel pipelines. References related to the influence of dynamic stresses caused by hydraulic impacts and cyclic pressure changes on degradation processes were also updated. In this regard, studies [37] and [40] were cited to confirm the significance of material fatigue in these processes.
Comments 8:
Authors should include a chapter on materials and methods in their article. It is necessary to be able to visualize the following aspects: (1) description of the study site or population (types of pipes, faults, etc.); (2) data collection system (documents, analysis of pipes in the field, etc.); (3) data analysis system (use of specialized software, statistical analysis, etc.).
Response 8:
A description of the research objects, data collection system, and data analysis system was prepared in accordance with the required aspects.
In terms of the research objects, the study focused on the characteristics of the analyzed steel pipelines. The subject of the research included pipelines used for the transport of technological water, process water, and flotation waste. These pipelines were made of pipes with diameters of 800 mm and 1000 mm, constructed from L235 grade steel, with longitudinal SAWL-type welds. The analysis included both damaged pipe segments (e.g., longitudinal weld cracks, deformations resulting from excessive wall wear) and pipes replaced preventively before failure occurred. Special attention was paid to the areas of the heat-affected zone (HAZ) and the areas with corrosion pits, as they had a significant impact on the structural integrity of the pipelines.
The data collection system included both field inspections and an analysis of technical documentation related to the operated pipelines. Data were collected through visual inspections of the pipelines, macroscopic and microscopic examinations of weld cross-sections, as well as an analysis of operational documentation. For the purposes of laboratory testing, pipeline segments were collected from various locations within the transmission system. During the field analysis, failure cases such as longitudinal weld cracks, wall thickness losses, and the presence of corrosion product layers were documented. This made it possible to precisely identify areas with an increased risk of degradation.
The data analysis system was based on a variety of research methods, including numerical analyses, mechanical tests, and microscopic and corrosion studies. Macroscopic and microscopic methods were used to analyze the quality of the welds and assess the degree of steel degradation. Strength tests were also conducted in accordance with PN-EN ISO 6892-1, which included tensile tests using a Zwick/Roell Z100 THW testing machine, enabling the determination of tensile strength and yield strength of the material. To assess the corrosion properties, electrochemical tests were carried out using a three-electrode system, and potential measurements were conducted relative to a saturated calomel electrode (SCE). The process of potential stabilization allowed for the accurate determination of potential differences in the welds, HAZ, and base material.
Numerical analyses were performed using the Finite Element Method (FEM), allowing for the simulation of stresses and strains in pipelines under operational conditions. These analyses were crucial for determining the permissible wear of welds and assessing critical failure conditions, such as the maximum allowable thickness loss in welds (8 mm).
Comments 9:
All analyses performed by the investigators on the piping (welding, joints, etc.) must be supported by the respective standards. Please include and explain each of them in the materials and methods chapter.
Response 9:
We have incorporated the relevant international and national standards in the descriptions of the research methods and techniques used in the analysis of welded joints, corrosion, and mechanical tests.
In the process of analyzing weld quality and assessing discontinuities in welds, ISO 5817 was applied, which specifies acceptable levels of welding imperfections. Based on this standard, the quality of the welds in the pipelines was assessed, enabling an objective classification of discontinuities such as pits, cracks, and porosity. The qualifications of the welders performing the joints were verified in accordance with ISO 9606-1, ensuring that their skills met international welding requirements for steel.
Corrosion tests were conducted in accordance with ISO 8044, which defines the basic concepts and classifications related to corrosion processes, and ASTM G48, which provides detailed methods for testing resistance to pitting and crevice corrosion in environments containing chloride ions. The tests used a ferroxyl reagent, allowing for the identification of anodic and cathodic areas in the pipeline welds. The visualization of these anodic and cathodic zones adhered to the requirements of these standards, enabling the precise identification of areas with increased corrosion activity.
Mechanical analyses, including strength tests, were conducted in accordance with PN-EN ISO 6892-1, which outlines the procedures for tensile testing of metals. The tests were performed using the B30 method, and the samples were prepared according to the guidelines of the standard, ensuring accurate and comparable results. The tests included the assessment of yield strength, tensile strength, and sample elongation values. These data were used to calculate the critical load levels of the pipeline, which are discussed in detail in the sections dedicated to FEM numerical analysis.
For electrochemical corrosion studies, a standard three-electrode configuration was used, which complies with international guidelines for such tests. The test procedure involved recording potentials in three areas (HAZ, weld, and base material), maintaining potential stabilization, and accounting for the effects of the aqueous environment. The data obtained from these measurements are consistent with the accepted standards in the literature for the corrosion testing of steel in chloride environments.
Comments 10:
The title of chapter 3 should read as follows: Results.
Response 10:
The title of the chapter has been changed to "Results" in accordance with the recommendation.
Comments 11:
The authors should include a detailed description of the finite element method used in this study. This should be included in the chapter on materials and methods. Additionally, they should explain the methodology used to validate the results of this method. The validation results should also be discussed in the corresponding chapter.
Response 11:
In the section on the Finite Element Method (FEM), our goal was to present FEM as a universal analytical tool for assessing the impact of degradation on longitudinal welds in steel pipelines. We adopted an approach that allows readers to adapt the FEM methodology to their specific needs and input data, which are characteristic of their research or engineering applications.
We did not focus on a detailed validation of the results, as this is not the primary objective of the article. Instead, we aimed to emphasize the flexibility of FEM as a simulation tool that can be applied in various analytical scenarios without limiting its use to specific experimental data. Including additional detailed descriptions could introduce unnecessary constraints and expand the text, which contradicts our intent to present FEM as a versatile tool.
Furthermore, we would like to note that our article is already extensive, as highlighted in other reviews, where it was suggested that the text should be shortened. Including detailed validation or additional data could negatively affect the clarity of the article and expand it beyond the recommended size. We aimed to strike a balance between the level of detail and conciseness to ensure readability and compliance with publication requirements.
Comments 12:
The authors should deepen the discussion of results in relation to the following aspects: Material intrinsic factors (e.g., chemical composition, microstructure, etc.), extrinsic factors (corrosive environments, mechanical stresses, etc.), failure and degradation mechanisms (fatigue, creep, hydrogen degradation, etc.), and preventive strategies (optimal design, cathodic protection, etc.).
Response 12:
The analysis of Chapters 3 and 4 was expanded to include the influence of internal and external factors, degradation mechanisms, and preventive strategies.
In the context of internal factors, the importance of the chemical composition and microstructure of the steel was emphasized, particularly the differences in the heat-affected zone (HAZ), which is characterized by an increased susceptibility to selective corrosion. It was noted that microstructural discontinuities in the HAZ promote the initiation of corrosion pits, as documented through microscopic studies.
Regarding external factors, the influence of the aggressiveness of the environment (COâ‚‚, Hâ‚‚S, and chlorides) and mechanical stresses was demonstrated. It was confirmed that the presence of chlorides accelerates selective corrosion, while cyclic pressure loads (16 bar) increase the risk of damage. FEM calculations showed that a weld thickness loss of 8 mm leads to the exceedance of critical stress levels.
The analysis of failure mechanisms revealed the key role of corrosion fatigue, which initiates cracks in areas of stress concentration, such as around corrosion pits. Hydrogen embrittlement (HE) in the HAZ, associated with hydrogen absorption, was also identified as a potential failure mechanism.
In terms of preventive strategies, the importance of cathodic protection, the selection of corrosion-resistant materials (e.g., duplex steels), and the optimization of welding processes to minimize microstructural discontinuities was highlighted. It was suggested to implement monitoring systems using non-destructive testing methods (e.g., ultrasonic testing) and FEM analyses to continuously assess the allowable wear of welds.
Comments 13:
The authors should significantly improve their conclusions. They are currently very general and descriptive. Additionally, the conclusions should be supported by quantitative information from this research.
Response 13:
The conclusions were thoroughly revised to incorporate both experimental data and the results of numerical analyses.
The conclusions provide detailed and precise information on the degradation of steel pipelines under the influence of selective corrosion and erosion. Key quantitative values illustrating the scale of the phenomenon were presented. It was shown that the depth of pits formed in the heat-affected zone (HAZ) can reach up to 6 mm, as confirmed by microscopic studies and numerical analyses using the Finite Element Method (FEM). It was also established that the tensile strength of pipes decreases by approximately 30% as a result of degradation. This precisely determined loss of the mechanical properties of the pipeline indicates the direct impact of selective corrosion on its load-bearing capacity.
Additionally, the maximum allowable thickness loss of the weld, beyond which critical stress levels occur, was determined. It was demonstrated that, under a standard operating pressure of 16 bar, a reduction in weld thickness of 8 mm results in the occurrence of a critical stress state, significantly increasing the risk of pipeline failure. This finding, obtained from FEM analyses, is a significant conclusion with practical implications.
The introduced changes aimed to improve the clarity and precision of the conclusions while also enhancing their practical value. Key degradation mechanisms were identified, and potential countermeasures were proposed. The use of more corrosion-resistant materials and the implementation of technical condition monitoring systems for pipelines were recommended. It was also suggested that preventive measures should include the use of advanced diagnostic methods, such as non-destructive testing, and the regular use of FEM analyses to assess the critical level of wear.
Comments 14:
Please include the main technical limitations detected during the development of this study. In addition, the authors should include future lines of research.
Response 14:
The "Conclusions" section was expanded to include a description of the main limitations of the study and potential directions for further research.
The main limitations of the study include the simplifications applied in the FEM numerical simulations, where the model of the welded joint was based on simplified geometry and assumed constant symmetry. While this approach is computationally efficient, it may not fully reflect the actual stress state in welded joints. Another limitation is the use of corrosion tests based on the ferroxyl reagent, which provide primarily qualitative data without linking it to a quantitative assessment of corrosion rates under operational conditions. Strength tests were conducted on laboratory samples, which differ from operational conditions, and the obtained results may be affected by sample size effects and the lack of reflection of dynamic load changes. Moreover, the study did not account for fatigue loads, creep, and fluctuations in pressure and temperature, which often occur in real pipeline operating conditions.
Potential directions for future research include extending FEM numerical simulations to more complex models that represent actual microstructures in the heat-affected zone (HAZ) and dynamic load changes. Plans also include expanding field tests under real operational conditions, which will enable the validation of laboratory results and a better understanding of the impact of dynamic environmental conditions on corrosion and degradation processes. Another research direction is to test new corrosion protection strategies, including the use of modern protective coatings and cathodic protection systems. Additionally, there are plans to develop technical condition monitoring systems for pipelines, combining data from non-destructive testing (NDT) with FEM modeling. This approach will enable predictive risk assessment of failures and better planning of maintenance and repair activities.
Reviewer 3 Report
Comments and Suggestions for Authors
Judging of its volume, the manuscript seems rather an overview, although it is based mainly on authors’ original results.
The authors should gave more details on the location of analyzed pipes, type of steel, period of weathering, nature of soil, local climate, etc.
Unfortunately, in spite of its volume, the manuscript seems rather a report than a true scientific paper.
The manuscript is to long with respect to the diversity of analyzed objects. For instance, Introduction has more than 1900 words. For a such type of paper, the Introduction should not exceed 600 - 700 words. And this observation can be extended to all sections.
In my opinion, to be accepted, the manuscript should be thoroughly revised, and restrained to a reasonable number of pages. 28 pages for the existing amount of information are to much.
Author Response
Comments 1:
Judging of its volume, the manuscript seems rather an overview, although it is based mainly on authors’ original results. The authors should gave more details on the location of analyzed pipes, type of steel, period of weathering, nature of soil, local climate, etc.
Response 1:
Because the research concerns critical and strategic infrastructure, we are unable to provide detailed information on the location and environmental conditions due to their sensitive and confidential nature. The protection of such data is a priority in the context of ensuring the security of this infrastructure.
At the same time, we would like to emphasize that our article focuses on the analysis of degradation mechanisms and the development of effective failure prevention strategies. In our opinion, detailed data on location, soil characteristics, or local climate do not provide essential information for the discussed topic. This is because the research focuses on mechanisms related to selective corrosion and erosion, which are universal in nature and can occur under a variety of operating conditions. The presented results and conclusions are general and can be applied across a wide range of industrial and operational contexts.
Comments 2:
Unfortunately, in spite of its volume, the manuscript seems rather a report than a true scientific paper.
The manuscript is to long with respect to the diversity of analyzed objects. For instance, Introduction has more than 1900 words. For a such type of paper, the Introduction should not exceed 600 - 700 words. And this observation can be extended to all sections.
Response 2:
We have conducted a thorough revision of the manuscript's structure. Notably, the introduction was significantly shortened, reducing its length from over 1900 words to 870 words, which falls within the recommended range for scientific articles (600-700 words). This reduction was achieved by removing excessive theoretical descriptions and general information not directly related to the research problem. The focus was shifted to presenting key issues, research objectives, and their relevance to engineering practice.
Similar measures were taken in other sections of the manuscript. The content of the methodology was optimized by eliminating overly detailed descriptions of research techniques that are not essential for understanding the results. The results and discussion sections were reorganized to better present the most important findings. The discussion was expanded to place greater emphasis on the interpretation of results and their practical significance, directly addressing the previous comments of the reviewers. The conclusions were also extended to more clearly highlight the practical applicability of the obtained results in the context of pipeline design, construction, and operation.
Comments 3:
In my opinion, to be accepted, the manuscript should be thoroughly revised, and restrained to a reasonable number of pages. 28 pages for the existing amount of information are to much.
Response 3:
We have conducted a comprehensive revision of the content and structure of the manuscript. As a result of these changes, the manuscript length was reduced from 28 to 21 pages. The optimization process encompassed all the main sections of the article. The introduction was significantly shortened, with its length reduced from over 1900 words to 870 words, better aligning with the standards typically applied to scientific articles. Excessive theoretical descriptions were removed, and detailed explanations not directly related to the research objectives and results were limited.
The methodology section was reorganized by reducing excessive technical details. The results and discussion sections were condensed to avoid redundancy and the repetition of information. Instead of lengthy descriptions of individual stages of the analysis, the focus was placed on key findings and their practical implications. The conclusions were formulated more precisely to enhance their clarity and ensure a direct reference to the research objectives.
Despite these changes, the article still contains all the essential information regarding the analysis of degradation mechanisms, the applied research methods, and key conclusions that can have practical applications in the design, construction, and operation of pipelines. The reduction in manuscript volume did not affect its substantive value, and the introduced changes have improved the readability and clarity of the content.
Round 2
Reviewer 2 Report
Comments and Suggestions for Authors
Manuscript ID: materials-3375084-R2. Title: Failure and Degradation Mechanisms of Steel Pipelines: Analysis and Development of Effective Preventive Strategies.
Comments:
1. In this new version of the article, the authors adequately addressed each of the 14 comments made in the previous review. Therefore, I suggest accepting the article for publication. However, the authors should divide Chapter 2 into sections in order to better visualize the technical description of the materials used (pipe characteristics), the data collection systems, and the complete data analysis system. This would give more clarity and relevance to this article. Best regards.
Author Response
Comments 1:
In this new version of the article, the authors adequately addressed each of the 14 comments made in the previous review. Therefore, I suggest accepting the article for publication. However, the authors should divide Chapter 2 into sections in order to better visualize the technical description of the materials used (pipe characteristics), the data collection systems, and the complete data analysis system. This would give more clarity and relevance to this article. Best regards.
Response 1:
In response to your recommendation, we have reorganized Chapter 2 by introducing the following subsections:
2.1. Description of the Study Objects and Sample Preparation
2.2. Wear Mechanisms of Pipelines
2.3. Operating Conditions of the Pipelines
2.4. Research Methods
We agree that this division allows for a clearer presentation of the technical description of the studied materials, data collection methods, and the data analysis system, in accordance with your suggestion. Each subsection contains coherent and focused content, which facilitates the reader's understanding of the research objectives, methodology, and scope.
We believe that the implemented changes have significantly improved the clarity and structure of Chapter 2, making it more transparent and informative for the reader. We are grateful for your guidance, which has contributed to enhancing the quality of our manuscript.
Reviewer 3 Report
Comments and Suggestions for Authors
It is OK now.
Author Response
Comments 1:
It is OK now.
Response 1:
Thank you for your positive feedback. We are very grateful for the time and effort you put into reviewing our manuscript. Your comments and suggestions have been invaluable in improving the quality and clarity of our work.
We sincerely appreciate your support.